# Transcutaneous Auricular Vagus Nerve Stimulation Alleviates Monobenzone-Induced Vitiligo in Mice

**DOI:** 10.3390/ijms25063411

**Published:** 2024-03-18

**Authors:** Shiqi Luo, Xinghua Meng, Jing Ai, Zhihong Zhang, Yanfeng Dai, Xiang Yu

**Affiliations:** 1State Key Laboratory of Digital Medical Engineering, School of Biomedical Engineering, Hainan University, Haikou 570228, China; shiqluo@hainanu.edu.cn (S.L.); xhmeng@hainanu.edu.cn (X.M.); jingai@hainanu.edu.cn (J.A.); zhzhang@hainanu.edu.cn (Z.Z.); 2Key Laboratory of Biomedical Engineering of Hainan Province, One Health Institute, Hainan University, Haikou 570228, China

**Keywords:** vitiligo, transcutaneous auricular vagus nerve stimulation (taVNS), monobenzone, oxidative stress

## Abstract

Vitiligo is a complex skin disorder that involves oxidative stress and inflammatory responses and currently lacks a definitive cure. Transcutaneous auricular vagus nerve stimulation (taVNS) is a noninvasive method for targeting the auricular branch of the vagus nerve and has gained widespread attention for potential intervention in the autonomic nervous system. Although previous research has suggested that vagus nerve stimulation can potentially inhibit inflammatory responses, its specific role and mechanisms in vitiligo treatment remain unknown. This study aimed to explore the therapeutic effects of taVNS in a mouse model of vitiligo induced by monobenzone. Initially, a quantitative assessment of the treatment effects on vitiligo mice was conducted using a scoring system, revealing that taVNS significantly alleviated symptoms, particularly by reducing the depigmented areas. Subsequent immunohistochemical analysis revealed the impact of taVNS treatment on melanocyte granules, mitigating pigment loss in the skin of monobenzone-induced vitiligo mice. Further analysis indicated that taVNS exerted its therapeutic effects through multiple mechanisms, including the regulation of oxidative stress, enhancement of antioxidant capacity, promotion of tyrosine synthesis, and suppression of inflammatory responses. The conclusions of this study not only emphasize the potential value of taVNS in vitiligo therapy, but also lay a foundation for future research into the mechanisms and clinical applications of taVNS.

## 1. Introduction

The vagus nerve is the longest and most widely distributed mixed nerve in the brain. It encompasses sensory fibers, special visceral fibers, and parasympathetic fibers [1]. By innervating various organs, such as the heart, lungs, spleen, pancreas, and liver, the vagus nerve plays a pivotal role in regulating essential processes like circulation, respiration, and digestion [2]. Furthermore, it projects into regions of the brain and the central nervous system (CNS), including the nucleus tractus solitarius (NTS), locus coeruleus (LC), thalamus, hippocampus, amygdala, and others [3]. Comprising a complex neuroendocrine immune network, the widespread distribution of the vagus nerve underscores its crucial role in systemic regulation, information transmission, and immune modulation [4].

Stimulation of the vagus nerve, particularly through transcutaneous auricular vagus nerve stimulation (taVNS), has emerged as an effective therapeutic approach. By avoiding the need for surgical implantation, taVNS, a noninvasive method, stimulates the vagus nerve through signals transmitted via its auricular branches, situated in the auricular concha and cavum concha [5]. taVNS has proven effective in treating various conditions, including depression, epilepsy, headache, tinnitus, atrial fibrillation, associative memory, schizophrenia, and pain [6,7]. Moreover, taVNS has shown promising results in studies related to Crohn’s disease and rheumatoid arthritis [8,9]. In the context of inflammatory diseases, taVNS significantly suppresses inflammatory responses, reduces the levels of inflammatory markers, and improves disease activity [10,11]. Vagus nerve stimulation unveils a novel neuroimmune paradigm, but much remains to be explored in terms of its applications and mechanisms across various diseases.

Vitiligo is a common skin disorder characterized by the loss of pigment-producing cells in the skin, resulting in the formation of depigmented symmetrical patches on the skin surface [12]. Although the exact pathogenesis of vitiligo remains unclear, it involves complex factors such as genetic predisposition, autoimmune responses, neural factors, oxidative stress, and melanocyte loss [13]. With a global prevalence of less than 1%, the development of vitiligo exhibits a certain degree of unpredictability among individuals. In addition to its aesthetic impact, vitiligo significantly affects the psychological well-being of patients [14]. Current treatment options for vitiligo, including phototherapy and medication, may lead to side effects that impact patients’ physical health. These side effects can range from mild skin irritation to more severe complications such as photosensitivity reactions or systemic effects like nausea and headache [15,16]. Additionally, vitiligo patients often experience relapses, even after successful prior treatments, and different treatment methods yield varying responses [17]. The limitations of existing treatments underscore the need for innovative approaches. Serving as a critical link between the nervous system and immune system, noninvasive electrical stimulation of the vagus nerve holds promise as a therapeutic approach for autoimmune diseases [18,19]. However, the potential therapeutic effects of taVNS in vitiligo remain largely unexplored.

Through long-term taVNS intervention, this research aimed to determine the potential ameliorative effects of vagus nerve stimulation on vitiligo. We employed a chemically induced vitiligo animal model induced by monobenzone, which mimics oxidative stress mechanisms in vitiligo, leading to lipid peroxidation and functional variations in melanocytes, ultimately enhancing autoimmune responses. Preliminary analysis of skin melanin changes through histopathological sectioning, examination of alterations in immune cells and cytokine levels, and measurement of oxidative stress levels collectively contributed to a comprehensive exploration of neuroimmune regulatory mechanisms. The uncharted territory of taVNS in the vitiligo treatment landscape can revolutionize the approach to treating vitiligo, offering a novel perspective for the treatment of autoimmune diseases.

## 2. Results

### 2.1. Physiological Effects of taVNS on HR

To assess the effectiveness of stimulating the auricular branch of the vagus nerve under our experimental conditions, we concentrated on stimulating the vagus nerve, which is predominantly distributed in the auricular cavity, as depicted in Figure 1A. To investigate the physiological effects of taVNS, we utilized a heart rate (HR) monitoring device for the precise recording of HR changes under a stimulation intensity of 20 Hz (Figure 1B). The results revealed a significant decrease in the HR of mice when taVNS was applied, signifying a pronounced regulatory effect on HR (Figure 1C). Compared to mice that did not receive taVNS, those subjected to stimulation exhibited a notable reduction in HR, providing a basis for subsequent investigations into the physiological phenomena and mechanisms (Figure 1D).

### 2.2. Intervention Effect of taVNS on the Progression of Vitiligo

Next, we investigated the potential of taVNS to intervene in the progression of vitiligo using a monobenzone-induced vitiligo mouse model. Following the establishment of the vitiligo model (Day 0), monobenzone was applied, and the mice in the vitiligo + taVNS group received taVNS treatment (Figure 2A). The extent of hair depigmentation in the back region of the mice was assessed through visual observation. As depicted in Figure 2B, compared with the normal group, the hair on the back of the mice showed obvious pigment loss after applying monobenzone to the designated area. However, after intervention, a decrease in pigment loss was observed in the taVNS group, and this phenomenon was further confirmed by statistical analysis. Statistical analysis of the depigmentation scores of the mice after modeling revealed that the decolorization score exceeded 2 points, indicating that the model was successfully established (Day 0). Importantly, the vitiligo group exhibited a significant increase in depigmentation score, while the group that received taVNS (vitiligo + taVNS) showed a notable decrease in score, highlighting the significant difference between the two groups (Figure 2C). Furthermore, we counted the white hair in the normal, vitiligo, and vitiligo + taVNS groups. The results indicated that the amount of white hair on the backs of mice in the vitiligo + taVNS group was significantly less than that in the vitiligo group (Figure 2D). To assess the histological changes in the skin tissue, we performed Masson–Fontana staining and HE staining to examine the back skin of mice with vitiligo, especially the changes in melanin granules. In Masson-Fontana staining, a silvery reaction occurs at the melanin, leading to the formation of black particles. After four weeks of intervention, the treated group showed a significant increase in the quantity of melanin around the hair follicles. Moreover, there was a substantial increase in the number of melanin granules that had migrated to the skin surface (Figure 3A). HE staining results demonstrated a noticeable lack of hair follicles in the skin of mice in the vitiligo group, accompanied by mild spongiotic changes and basal cell vacuolization. In contrast, the skin exhibited significant improvement after treatment, characterized by an increase in hair follicles and a reduction in pathological alterations (Figure 3B). These findings collectively suggest that taVNS has the potential to intervene in the progression of vitiligo in mice, as evidenced by the reduced depigmentation and favorable histological changes observed in the treated group.

### 2.3. Impact of taVNS on Oxidative Stress and Melanocyte Apoptosis in Mice

Recent studies have suggested a link between the progression of vitiligo and oxidative stress [20]. In this study, we assessed the concentration of malondialdehyde (MDA) and the activity of superoxide dismutase (SOD) and catalase (CAT) in the skin tissues of mice following the intervention. As shown in Figure 4A–C, oxidative stress in the skin led to a sharp decrease in SOD and CAT levels and a significant increase in MDA levels in the vitiligo group. However, after taVNS, the MDA concentration significantly decreased, and SOD and CAT activities were restored (*p* < 0.05). Oxidative stress can induce apoptosis of melanocytes in the skin, resulting in reduced tyrosinase synthesis in apoptotic melanocytes [21,22]. We measured the level of tyrosinase in mouse skin, and the results showed a significant reduction in tyrosinase content in the skin of vitiligo mice. After taVNS intervention, there was a significant increase in tyrosinase synthesis in the mouse skin (Figure 4D,E), indicating that taVNS may increase melanin granule formation by promoting tyrosinase synthesis. Taken together, these findings suggest that taVNS intervention significantly reduces oxidative stress and promotes melanin synthesis in the skin of vitiligo mice.

### 2.4. Anti-Inflammatory Capacity of taVNS in Vitiligo Mice

Inflammatory factor levels are crucial indicators that are typically elevated in patients, reflecting an intensified inflammatory state [23]. Therefore, measuring the levels of inflammatory factors is a common method for assessing the progression and therapeutic efficacy of vitiligo treatment. As illustrated in Figure 5A–C, the serum of monobenzone-induced vitiligo mice exhibited a marked increase in proinflammatory cytokines, such as TNF-α, IFN-γ, and IL-6. However, there was a significant reduction in the levels of these cytokines after taVNS treatment. Furthermore, we performed immunofluorescence staining of the skin in the depigmented area of the mice. The results indicated that the expression of inflammatory factors was notably greater in the skin of the vitiligo group than in that of the other two groups (Figure 5D,E). These results underscore the effectiveness of the taVNS intervention in suppressing the inflammatory response within the body.

### 2.5. Effect of taVNS on CD8^+^ T Cell Infiltration

Studies have indicated that during the development of vitiligo, CD8^+^ T cells are recruited to the dermis and contribute to the destruction of melanocytes, potentially affecting melanin production and representing one of the possible factors underlying vitiligo pathogenesis [24,25]. To explore the relationship between taVNS and immune infiltration, we conducted an in-depth investigation into CD8^+^ T cell infiltration. In this study, we conducted immunofluorescence staining of the skin in the depigmented area to assess CD8^+^ T cell infiltration. It was evident that the number of CD8^+^ T cells within the skin of the vitiligo group was greater than that within the skin of the other two groups (Figure 6A). Furthermore, we performed flow cytometry analysis of skin from the depigmented area, and our gating strategy is depicted in Figure 6B. The results revealed a significant increase in CD3^+^ CD8^+^ T cells in the dermal skin of vitiligo mice compared to that in the dermal skin of normal mice. However, after taVNS intervention, the percentage of CD3^+^ CD8^+^ T cells significantly decreased (Figure 6C). These findings suggest that taVNS intervention markedly reduces CD8^+^ T cell infiltration in the depigmented area, potentially alleviating melanocyte damage caused by CD8^+^ T cells.

## 3. Discussion

In this study, we utilized the monobenzone-induced mouse model of vitiligo to investigate the impact of taVNS, providing crucial insights into the regulatory mechanisms of the vagus nerve system in the development of vitiligo. First, we observed the therapeutic potential of taVNS in a mouse model of vitiligo, as reflected by the alleviation of disease manifestations. Furthermore, we explored the effects of taVNS on the antioxidant capacity and apoptosis of melanocytes in mice. Our experimental results indicate that vagus nerve system stimulation has a certain regulatory effect on both aspects, possibly associated with the neural regulatory mechanisms involved in maintaining redox balance and cell apoptosis. These findings suggest the significant physiological role of the vagus nerve system in maintaining cellular homeostasis. Concurrently, vagus nerve stimulation has been shown to reduce inflammatory factor levels, and in our study, we also measured inflammatory marker levels in vitiligo mice, revealing a moderate reduction after vagus nerve stimulation intervention. Finally, we investigated the influence of taVNS on the infiltration of CD8^+^ T cells in mouse skin and found that the vagus nerve system may modulate the development of vitiligo by affecting the infiltration of immune cells. Further mechanistic studies could help elucidate the exact role of the vagus nerve system in this process. Overall, our experimental results indicate that taVNS can regulate the development of vitiligo in mice through multiple pathways (Figure 7).

In previous studies, vagus nerve stimulation has been demonstrated to play a significant regulatory role in the inflammatory process [26,27,28]. In our research, we further explored the potential impact of vagus nerve stimulation in a mouse model of vitiligo. We observed a significant decrease in inflammation in vitiligo mice after vagus nerve stimulation, consistent with previous findings in inflammatory diseases, suggesting that the vagus nerve system may alleviate tissue damage and the development of related inflammatory diseases by inhibiting inflammatory reactions [9,29,30,31,32]. However, unlike previous studies that have focused mainly on the regulation of inflammatory processes, our study explored the potential effects of vagus nerve stimulation on skin diseases. Specifically, we emphasized the impact of taVNS on the infiltration of CD8^+^ T cells in mouse skin. Taken together, our results indicate that taVNS modulates immune cell infiltration in the skin, which is crucial for the development of skin diseases such as vitiligo. While previous research has reported the regulatory role of the vagus nerve system in the inflammatory process, limited studies have investigated the detailed regulation of immune responses in skin diseases. Our research findings provide preliminary evidence for a new role of the vagus nerve system in skin immune regulation, particularly in the modulation of immune cell infiltration.

The onset of vitiligo involved intricate interactions between melanocytes and hair follicles. In mice, melanocytes are predominantly located within the hair follicles. Some studies have suggested that the origin of melanocytes and hair follicles could be traced back to neural crest cells [33]. It was worth noting that the skin was primarily innervated by sympathetic nerve fibers and also influenced by autonomic nerve fibers. These nerve fibers innervated the dermis, blood vessels, and hair follicles, allowing for neurons to connect to the follicles and thereby influence their function and the physiological activities of the skin [34,35]. In addition, we observed a significant increase in hair follicle proliferation in the skin after taVNS treatment. Existing studies have suggested that activation of the PI3K pathway promotes the proliferation and differentiation of hair follicle stem cells, thus driving hair follicle regeneration [36,37]. Interestingly, vagus nerve stimulation could influence the activity of the PI3K pathway by activating the cholinergic pathway [38,39]. Therefore, vagus nerve stimulation may indirectly affect the activity of the PI3K pathway, thereby impacting hair follicle regeneration. This discovery not only expands our understanding of the scope of the actions of the vagus nerve system but also opens new directions for exploring its potential applications in the treatment of skin diseases. However, it should be noted that further in-depth research is required to elucidate the exact mechanisms of action of the vagus nerve system in skin diseases. Future experiments and clinical studies will contribute to revealing the specific regulatory mechanisms of the vagus nerve system in different skin diseases, providing additional comprehensive theoretical support for its potential applications in skin disease treatment.

The monobenzone-induced vitiligo model primarily focuses on melanocyte apoptosis and is well suited for studying molecular and cellular mechanisms related to this process [40,41,42]. However, compared to models induced by autoimmune mechanisms, there is an obvious lack of autoimmune components attacking melanocytes in the monobenzone-induced model, potentially leading to fully replicating the complexity of immune abnormalities observed in vitiligo patients [43,44,45]. Additionally, vitiligo is a heterogeneous disease, and different patients may have different pathogenic mechanisms [46]. The monobenzone-induced model may not fully account for this heterogeneity. However, evaluating the therapeutic effects of auricular vagus nerve stimulation in vitiligo may require assessment in additional models.

Up to now, the exact pathogenic factors of vitiligo have remained unclear but may involve multiple factors, including immune system abnormalities, genetic factors, and environmental triggers. Clinically, various treatment methods are available depending on individual patient characteristics [47]. These include phototherapy, oral corticosteroids, immunosuppressive drug therapy, and laser therapy, among others [48]. This study introduces taVNS as a noninvasive treatment method for the first time in a mouse model of vitiligo induced by monobenzone treatment. Considering the association of the vagus nerve system with immune responses, antioxidation, anti-inflammation, and various physiological processes, taVNS may yield better therapeutic effects in the treatment of vitiligo. Taken together, our study results indicate that taVNS may have therapeutic effects by regulating antioxidant capacity, inhibiting melanocyte apoptosis, and influencing anti-inflammatory capabilities, suggesting that it is involved in the multifaceted regulation of vitiligo. By affecting immune cell infiltration, taVNS may help to reduce the attack of the immune system on melanocytes, counteracting the development of vitiligo. In addition, previous studies also have reported the successful construction of a chronic stress-induced vitiligo mouse model based on the monobenzone-induced model, in which psychological induction further suppressed melanocyte generation and affected mouse behavior [49,50]. The comprehensive regulation of taVNS may better align with the complex pathophysiological mechanisms of vitiligo, encompassing immune system abnormalities, melanocyte apoptosis, and psychological stress.

In summary, taVNS intervention can alleviate the development of vitiligo through the regulation of multiple physiological processes, highlighting its potential clinical applications in vitiligo treatment. Moreover, elucidation of the specific treatment mechanisms will provide valuable clues for future in-depth exploration of the potential role of the vagus nerve system in autoimmune disease.

## 4. Materials and Methods

### 4.1. taVNS and Physiological Monitoring

Six-week-old male C57BL/6 mice were purchased from Changsha Hunan Silaike Jingda Laboratory Animal Co., Ltd. (Changsha, Hunan, China). These mice were maintained by the Animal Experimental Center at Hainan University. The animals were housed in a temperature-controlled environment, with temperatures maintained within the range of 20–25 °C and subjected to a standard 12 h light and 12 h dark cycle. Throughout the experimental process, the mice had unrestricted access to food and water.

To perform taVNS, the mice were anesthetized with 2.5% isoflurane and maintained with 1.5% isoflurane. In this process, the vagus nerve of mouse ear was stimulated by transcutaneous electrical stimulation using (Hwato SDZ-II, Suzhou, China) electronic acupuncture instrument. In order to ensure stability, the positive and negative electrodes of the electronic acupuncture instrument are connected with the acupuncture needle to stimulate the mouse auricle area. The stimulation parameters were set as a continuous wave with a frequency of 20 Hz and a stimulation width of 0.2 ms.

HR monitoring: under 2.5% induction and 1.5% maintenance of isoflurane anesthesia, the mice were placed in the supine position on a heating pad. Subsequently, the electrocardiogram (ECG) electrode module was connected to the mice to record ECG signals and measure HR. Following ECG electrode placement, HR signals were amplified using a biological amplifier (Bio-Amp Octal, ADInstruments, Colorado Springs, CO, USA) and digitized using PowerLab (ADInstruments, Colorado Springs, CO, USA). The heart rate baseline needed to be carefully adjusted and recorded before stimulation. Finally, the digital signals were transmitted to a computer running LabChart8 (ADInstruments, Colorado Springs, CO, USA) for recording, analysis, and storage, ensuring the accuracy and reliability of the experimental data.

### 4.2. Induction of a Vitiligo Model in Mice Using Monobenzone

The vitiligo mouse model was established by applying 40% monobenzone cream; the induction of vitiligo in mice involved the following steps: Initially, hairs were removed from the 2 × 2 cm area of the back of the mice. On the first day following depilation, a 40% monobenzone solution (50 mg per day, Xinzhuangyan, Nanjing, China) was continuously applied to the specified area for 45 days.

### 4.3. taVNS Treatment

Mice were divided into 3 groups composed of 4 individuals: (i) normal, (ii) vitiligo (untreated), and (iii) vitiligo + taVNS. To maintain the vitiligo status of the mice, all mice were treated with monobenzone daily during the treatment period. The mice in the untreated group were anesthetized at the same time every day without stimulation, while the mice in the taVNS group received stimulation via the auricular vagus nerve for 30 min per day, and the stimulation parameters were consistent with those used previously. This intervention lasted for 4 weeks. Subsequently, skin lesions were scored using a bleaching scoring method, and skin tissues and serum were collected on the last day of treatment for further analysis.

Vitiligo model evaluation: the vitiligo model was evaluated using a scoring system based on the extent of depigmentation in the treated area. A score of 1 indicated an area less than 10% depigmentation, 2 indicated 10–25%, 3 indicated 25–50%, 4 indicated 50–75%, and 5 indicated more than 75% depigmentation.

Manual white hair counting: white hair counting was performed by visually inspecting the dorsal region of each mouse and tallying the number of white hair clusters present.

These additional methods were employed to assess the effectiveness of the vitiligo induction and to quantify the extent of depigmentation and white hair appearance in the experimental mice.

### 4.4. Histological Analysis

Skin tissues from the depigmented areas were fixed in 4% buffered paraformaldehyde, dehydrated in a series of ethanol and xylene concentrations (75%, 85%, 95%, and 100%), and embedded in paraffin. The processed tissues were then sectioned into 5 μm slices. These sections were subjected to HE and Masson–Fontana staining to determine melanin content accurately. These staining schemes lay a reliable foundation for further exploration of the organizational structure, which is conducive to the preliminary evaluation of melanin content and related structures.

### 4.5. Flow Cytometry

Skin samples were collected from the central 1 × 1 cm area of the back, cut into small pieces, and placed in a pre-configured solution of RPMI-1640 and 0.33 mg/mL Liberase TL enzyme mixture (Roche, Mannheim, Germany) for efficient tissue digestion. The skin tissue was transferred to a 1.5 mL centrifuge tube, cut as finely as possible with scissors, and incubated in a shaking bath or incubator at 37 °C for 60 min with 500–1000 µL of enzyme digestion solution. After thorough grinding, the cell suspension became misty, and was filtered through a 70 μm cell strainer. Subsequently, the digested tissue was resuspended in PBS, centrifuged, and filtered through a cell strainer, resulting in a single-cell suspension.

The collected skin single-cell suspensions were subsequently transferred to labeled flow tubes and subjected to immunocellular detection using BV605-CD45 (563053, BD Biosciences, Franklin Lakes, NJ, USA, 1:200) for immune cell detection and APC/Cy7 Anti-Mouse CD3e (100329, BioLegend, San Diego, CA, USA, 1:200) and Alexa Fluor 647 anti-mouse CD8a (100724, BioLegend, San Diego, CA, USA, 1:200) for CD8^+^ T cell detection. Fixable Viability Dye eFluorTM 506 was used to exclude dead cells, and all the antibodies were incubated on ice for 30 min. Finally, cell analysis and sorting were performed using a CytoFLEX flow cytometer, and the data were further analyzed with CytExpert 2.4 analysis software to ensure accurate detection and analysis of the cells. This procedure provides an efficient and accurate means for the in-depth study of skin cells.

### 4.6. Assessment of Antioxidant Stress and Anti-Inflammatory Activity in the Skin Tissues of Vitiligo Mice

After taVNS treatment, we collected mouse skin tissues and used the extraction solution provided with the kit to obtain tissue homogenates and collected the supernatant. Subsequently, we quantified the activities of superoxide dismutase (SOD) and catalase (CAT), as well as the concentration of malondialdehyde (MDA), using commercial assay kits to assess the level of antioxidant stress. The CAT and SOD activities were measured by using an SOD Activity Assay Kit (BC0170, Solarbio, Beijing, China) and CAT Activity Assay Kit (BC0205, Solarbio, Beijing, China), respectively. The MDA concentration was measured by using an MDA content assay kit (BC0025, Solarbio, Beijing, China) according to the protocol provided by the manufacturer.

Simultaneously, mouse blood was collected by centrifugation (4 °C, 3000 rpm, 15 min), after which the serum was collected. Then, following the recommended procedures provided by the manufacturer, we used commercial ELISA assay kits purchased from JINGMEI Biotechnology to measure the levels of tumor necrosis factor-α (TNF-α), interferon-γ (IFN-γ), and interleukin-6 (IL-6) in mouse serum to evaluate their anti-inflammatory activity.

### 4.7. Immunofluorescence Staining

To conduct immunofluorescence analysis of the skin tissues, a small section of skin was removed from the depigmented area on the back of the mice. The skin tissue was fixed in 4% paraformaldehyde at 4 °C for 24 h, followed by dehydration in a 30% sucrose solution at 4 °C. Subsequently, the tissue was frozen in Optimal Cutting Temperature (OCT) compound and sectioned into 15 μm slices using a cryostat (RWD FS800A, Shenzhen, China).

The slices were washed three times with PBS to remove OCT and then immunostained with Alexa Fluor 647 CD8a (100724, BioLegend, San Diego, CA, USA, 1:200) to detect CD8^+^ T cells in the skin tissue. Additionally, immunostaining was performed using a TNF Alpha Polyclonal antibody (17590-1-AP, Proteintech, Chicago, IL, USA, 1:200), an IFN Gamma Polyclonal antibody (15365-1-AP, Proteintech, Chicago, IL, USA, 1:200), and a Goat Anti Rabbit IgG (H&L)-Alexa Fluor 594 (RS3611, Immunoway, Plano, TX, USA, 1:200) to assess the levels of inflammatory factors in the skin tissue.

All the sections were imaged with a laser scanning confocal microscope (Olympus FV3000, Tokyo, Japan). Finally, ImageJ2 software was used to analyze the acquired data, which provided quantitative information on the fluorescence signals related to CD8^+^ T cells and inflammatory factors.

### 4.8. Western Blotting (WB) Analysis

The cells in the skin tissue were lysed using RIPA lysis buffer. The lysed samples were then centrifuged at 8000× *g* for 10 min at 4 °C, after which the resulting supernatant was collected. Protein quantification was performed using a BCA protein assay kit (P0010S, Beyotime, Nantong, China) following the instructions provided with the kit.

Twenty-five micrograms of total protein were loaded onto a 10% SDS-PAGE gel, followed by protein separation through electrophoresis. Subsequently, the separated proteins were transferred onto a polyvinylidene fluoride (PVDF) membrane. Immunoblotting utilized the following antibody: anti-tyrosinase antibodies (WL01000, Wanleibio, Shenyang, China), diluted at 1:500. The secondary antibody was anti-rabbit horseradish peroxidase (HRP) conjugates (A0208, Beyotime, Nantong, China), diluted at 1:1000. Following immunoblotting, chemiluminescence detection was employed for membrane imaging, and exposure was conducted in a fluorescence and chemiluminescence gel imaging system (Cytiva Amersham 800, Cytiva, Amersham, UK) to visualize protein bands. β-actin (GB15001, Servicebio, Wuhan, China) served as the as the internal reference protein. Images were analyzed using ImageJ2 software for data interpretation.

### 4.9. Statistical Analysis

All the data are presented as the mean ± SD from at least three independent experiments. One-way ANOVA was used for multiple group comparisons, and Student’s *t*-test (two-tailed) was used for comparisons of two groups. Statistical analyses were performed using GraphPad Prism software version 8. A *p*-value of <0.05 denoted statistical significance.

## Figures and Tables

**Figure 1 ijms-25-03411-f001:**
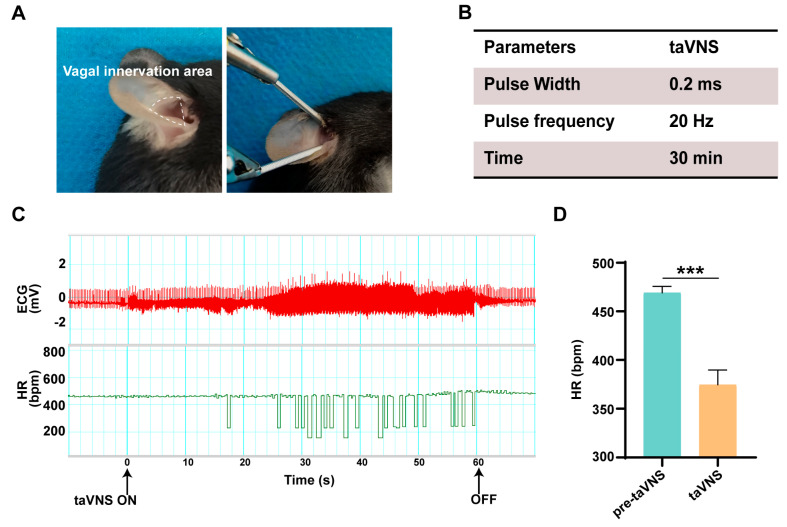
Physiological effects of taVNS on HR. (**A**) Electrical stimulation applied within the innervation area of the vagus nerve in the ear. (**B**) The taVNS stimulation parameters used in this study. (**C**) Representative results of the changes in ECG (red trace) and HR (green trace) elicited by a series of taVNS. (**D**) Counting the mean value of HR in the pre-taVNS and taVNS. Data are presented as the mean ± SD; differences between two groups were analyzed by Student’s *t* tests. *** *p* < 0.001.

**Figure 2 ijms-25-03411-f002:**
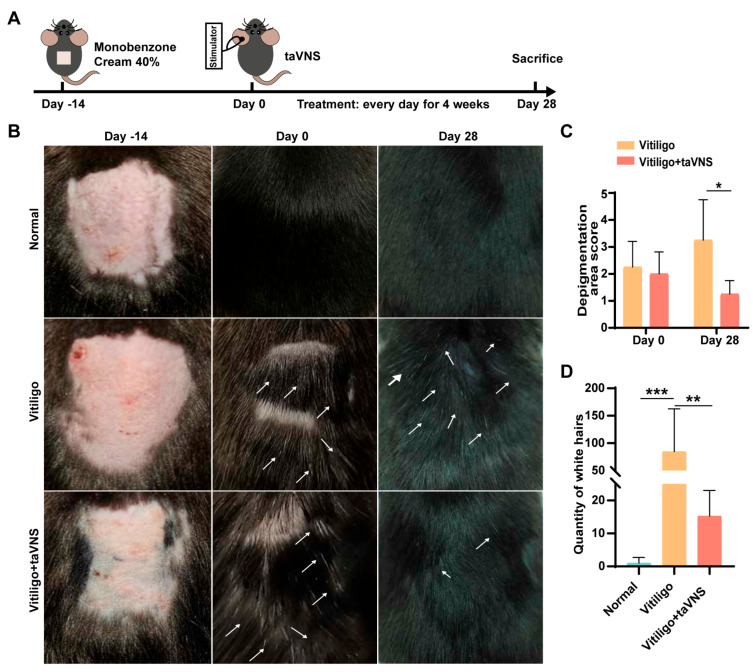
Intervention effect of taVNS on progression in vitiligo. (**A**) Schematic illustration of the animal models and treatment schematics. (**B**) Monitoring of back decolorization of the mice. The white arrow indicates the white hair area. (**C**) Depigmentation scores in mice with vitiligo. (**D**) Statistics for the number of white hairs after taVNS treatment. Data are presented as the mean ± SD (n = 4); Student’s *t*-test was used for comparing two groups, and one-way ANOVA was used for multiple groups. * *p* < 0.05, ** *p* < 0.01, *** *p* < 0.001.

**Figure 3 ijms-25-03411-f003:**
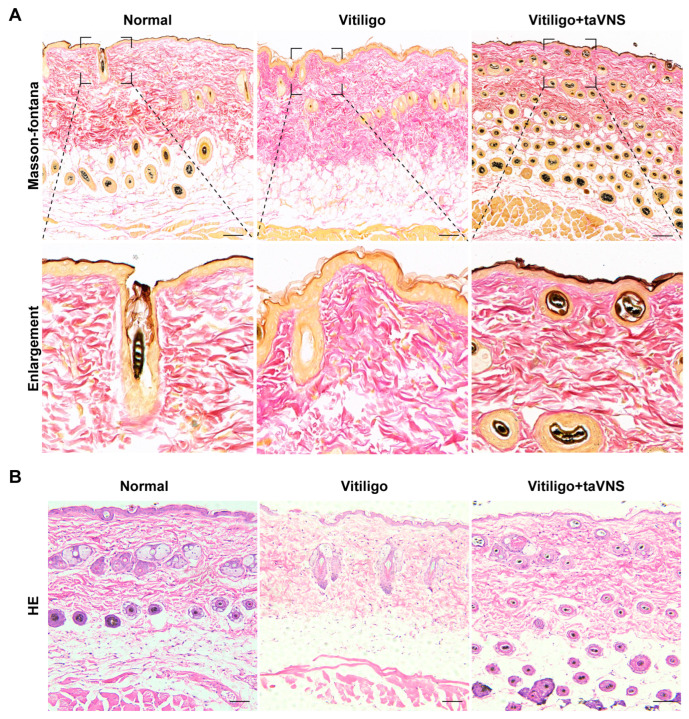
HE and Masson–Fontana staining of depigmentation skin after taVNS. (**A**) Masson–Fontana staining of skin tissue, magnification of the boxed area is shown below. Magnification: 100×. (**B**) HE staining of skin tissue. Magnification: 100×. Representative images are shown. Scale bar: 100 µm.

**Figure 4 ijms-25-03411-f004:**
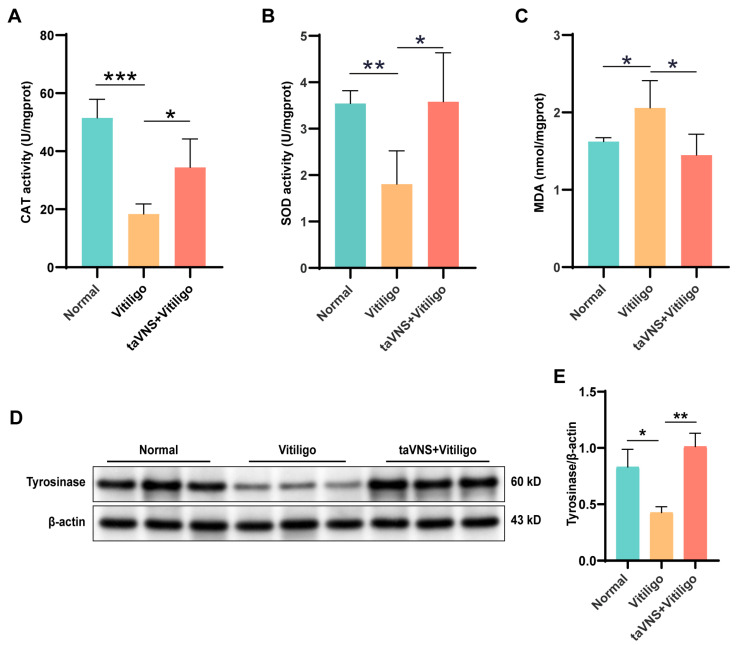
Impact of taVNS on oxidative stress and melanocyte apoptosis in mice. (**A**–**C**) CAT activity, SOD activity, and MDA content in mouse skin tissues after taVNS treatment for 28 days. (**D**) Protein expression levels of tyrosinase in skin tissues. (**E**) Quantification of the amount of tyrosinase relative to β-actin in the skin. Data are presented as the mean ± SD; multiple groups were tested using one-way ANOVA. * *p* < 0.05, ** *p* < 0.01, *** *p* < 0.001.

**Figure 5 ijms-25-03411-f005:**
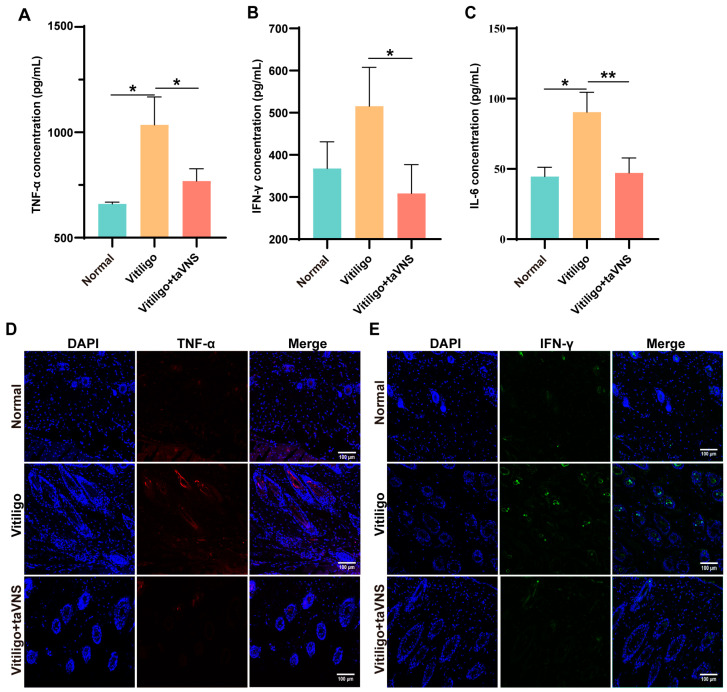
Anti-inflammatory capacity of taVNS in vitiligo mice. (**A**) TNF-α, (**B**) IFN-γ, and (**C**) IL-6 concentrations in mouse serum after taVNS treatment for 28 days. (**D**,**E**) Immunofluorescence images of skin inflammatory factors in different groups of mice (blue: DAPI; red: TNF-α; green: IFN-γ). Scale bar: 100 µm. Data are presented as the mean ± SD (n = 4); multiple groups were tested using one-way ANOVA. * *p* < 0.05, ** *p* < 0.01.

**Figure 6 ijms-25-03411-f006:**
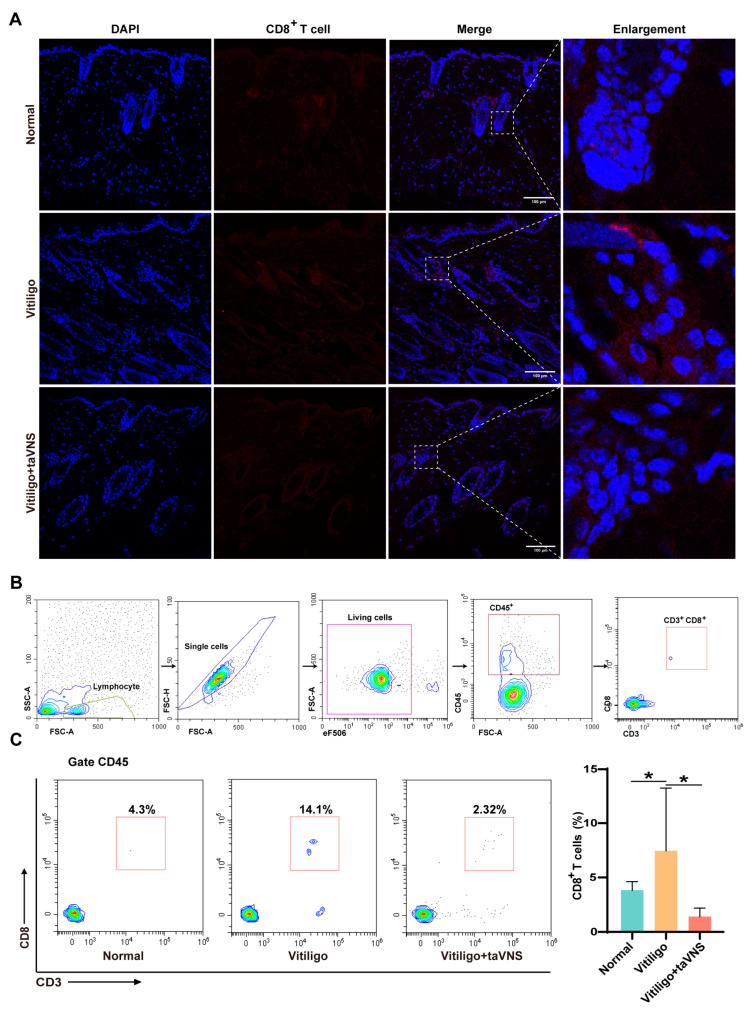
Effect of taVNS on CD8^+^ T cell infiltration. (**A**) Immunofluorescence images of skin CD8^+^ T cells after taVNS treatment for 28 days, enlarged view of the boxed area is shown on the right side. (**B**) The gating strategy for flow cytometry analysis. (**C**) Representative flow cytometry plots and quantification of CD3^+^ CD8^+^ T cells (pregated on CD45^+^ live singlets). Scale bar: 100 µm. Data are presented as the mean ± SD; multiple groups were tested using one-way ANOVA. * *p* < 0.05.

**Figure 7 ijms-25-03411-f007:**
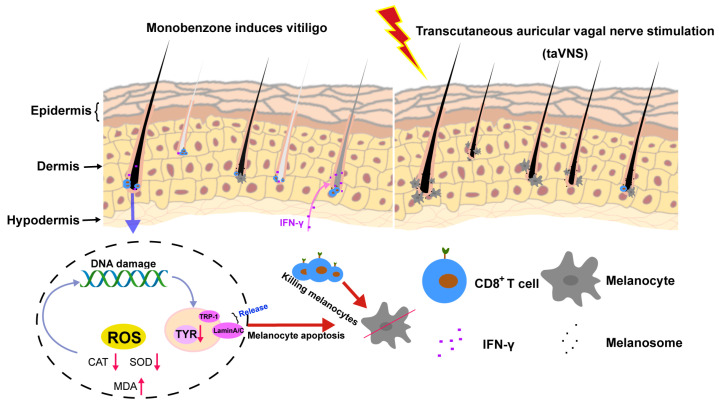
A schematic diagram of the mechanism of taVNS intervention in mouse vitiligo. The mechanism of this intervention involves the modulation of oxidative stress, restoration of antioxidant capacity, promotion of tyrosine synthesis, and suppression of the inflammatory response. Intervention effectively reduces depigmentation, promotes the formation of melanin granules, and contributes to favorable histological changes in the skin of vitiligo mice.

## Data Availability

All the data used to support the findings of this study are included within this article.

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
