# Peer review of "Transcutaneous Auricular Vagus Nerve Stimulation Alleviates Monobenzone-Induced Vitiligo in Mice"

_ijms, 2024, doi:10.3390/ijms25063411_

Round 1

Reviewer 1 Report

Comments and Suggestions for Authors

This paper seeks to measure the effects of taVNS on a mouse model with induced vitiligo. While vitiligo does not affect a very significant proportion of the population, a noninvasive treatment could be useful in improving their way of life, as well as understanding the autoimmune factors behind flare-ups of this disorder.

Do you have any justification for this small sample size? 4 per group is very small.

Figure 1C: what is the time scale for this? How can you be sure that those drops in HR are related to stimulation as opposed to issues with the LabChart algorithm mistaking noise artifact for ECG? The baseline HR does not seem to change, but those sudden drops might be related to stimulation artifact.

There are crucial limitations to the monobenzene-induced model, but they are addressed properly in the discussions. I do agree with the authors that additional models are necessary to evaluate these taVNS effects.

Section 4.1: More details on the how long animals were under anesthesia, how long was stimulation. This section also requires justification for both frequency and amplitude. What amplitudes were used? Was it the same for all animals or different, and how was this determined? I see some details in 4.3, but not all. Consider editing titles of these sections so they do not repeat and contain the relevant information.

No methods section regarding counting white hairs. How was this done?

Reviewer 2 Report

Comments and Suggestions for Authors

The manuscript by Luo, S. et al. describing a possible treatment of vitiligo in a mouse model through vagus nerve stimulation seems complete and scientifically sound. However, there are 2 major issues that need solution.

The first is regarding the resolution and the size of the employed photographs, which is very low. You need to heavily improve this point, may be if you divide and / or enlarge the images may help. Moreover, histological images are not representative (see minor comments).

Another major comment is respecting the protocol for immunofluorescence staining. I don´t know if it is a drafting error, but if you fix the tissue with formaldehyde, the immunofluorescence is not reliable at all. For this reason, this material is commonly directly frozen in OCT. If you fix the sample in formaldehyde (a complete day!) and then you freeze it, antigenicity may become voided. This may explain the low antigenicity observed in the immunofluorescence images.

The language is fine and I can´t find relevant issues.

There are various minor comments:

- Introduction, page 2, lines 60-62. I think this sentence needs at least one supporting reference.

- Introduction, page 2, line 70. I think you refer here to HISTOpathological sectioning (…).

- Results, page 2, lines 77-80. These sentences are not results. They can be part of the introduction section or the discussion section. In addition, it´s commonly accepted to merge discussion and results in scientific literature.

- Results, page 2, lines 106-107. This sentence is confusing, are you referring to the taVNS group? Moreover, you should be more specific with the statistical analysis results.  

- Results, page 2, line 115. Immunohistochemistry?? Where are the immunohistochemical images??? You do not refer to any image in this sentence. This one should be changed or removed. To my knowledge, melanosomes are the organelles where the melanin is synthetized (in the melanocytes, obviously), and are not necessarily stained with Masson-Fontana, in contrast with melanin granules. However, an interesting article in this purpose may be the one by Yoshikawa-Murakami et al. in Int J Mol Sci in 2020: 21(22):8514. HMB45 is a common marker of melanosomes in clinical practice.

- Results, page 2, line 119. You haven´t mentioned at any point that Masson Fontana stains (in black) the melanin granules. In the provided image (I guess you actually refer to Figure 2D), the skin surface (epidermis) can be hardly imagined. The detailed “Enlargement” insert below the Masson Fontana line only depicts hair follicles that can be seen in the low magnification images. What I can see in this D part of Figure 4 is that the vitiligo specimen has very few hair follicles and that hair follicles in this case are only located in the dermis, leaving the hypodermis without follicles. Another clear finding in this figure is that taVNS induces in the vitiligo specimen a strong hair follicle hyperplasia. However, it also appears that normal and taVNS specimens have stronger pigmentation in hair follicles and epidermis.

- Results, page 2, lines 119-121. It´s very difficult, even for a trained specialist, to accurately differentiate melanocytes in HE slides (you can differentiate some of them, but not all). In any case, the text in the results section do not correspond to the provided images of Figure 2E (I also guess here you actually refer to Figure 2E), you should include high power images from the epidermis to support the provided affirmation. By the way, what´s evident with the HE staining is the same findings I mentioned in the Masson-Fontana one regarding the distribution and number of hair follicles in dermis and hypodermis.

- Results, Figure 2. In (B) I can´t hardly see any difference between the different groups; in addition, the image resolution is quite low and I can´t zoom to see what the arrows indicate. Arrows are not mentioned in the figure legend.

- Results, line 178. T cells are recruited FROM the dermis?

- Results, line 186. Percent?

- Results, Figure 5. Again, you should explain the meaning of the arrows in 5A. At this magnification and resolution, I can see nothing in this image. Nothing.

- Results, Figure 6. This figure appears quite exaggeraged. The figure depicts 80% of white hairs in your model of vitiligo, which is not what is depicted in the real model of Figure 2B

- Discussion, page 8, line 241. Will contribute to reveal the specific...

- Materials and methods, page 9, line 282. How many mice? 12??

- Materials and methods, page 9, line 298. How many 6-week-old male C56BL mice? 8???

- Materials and methods, page 10, lines 325-327. HE does not render “precise assessment of melanin content and related structures”. HE gives an initial approach to tissue. HE can be complemented with special (histological) techniques, immunohistochemistry, electron microscopy, etc. in order to reach the precise assessment of whatever you want. For example, immunohistochemistry against HMB45 is a common marker of melanosomes.

- Materials and methods, page 11, line 368. What´s subcutaneous mucosa? The epidermis is an stratified epithelium with no mucus. The dermis is connective tissue, with no epithelium. Cutaneous adnexa are glandular structures with no mucus and really difficult to remove. Mucosa is a mucus producing epithelium, for example the colon.

Reviewer 3 Report

Comments and Suggestions for Authors

Overview of the manuscript

The manuscript focuses on studying the involvement of vagus nerve stimulation in relieving the vitiligo symptoms. The authors use the transcutaneous auricular vagus nerve stimulation (taVNS) in a mice model of monobenzone-induced vitiligo. To delve deeper into the topic of the work, the symptom scoring system, immunohistochemical analysis and the study of inflammatory factors expression were used. The authors found that taVNS exerts its therapeutic effects regulating oxidative stress, enhancing antioxidant capacity and suppressing inflammatory responses. The authors conclude on the importance of future explorations on the mechanisms and clinical applications of taVNS.

 GENERAL  COMMENT

The work is interesting and paves the way for a broad consideration of transcutaneal Vagus nerve stimulation in several disease. The experimental plan is accurate and well performed. The investigation methodologies are adequate to support the results and discussion. In same places the manuscript should be better written, and figures better presented.

Specific comments

Introduction

Pag. 2, line 57: better explain the issue of medication side effect and add references.

 Results

Pag. 2. Line 77-79: this paragraph repeats arguments of introduction, it is useless here. Remove it.

Pag. 2, Line 89-90: the sentence is a repetition of arguments presented in introduction. Remove it.

Pag. 2, line 119-123: the indication of figure 4 is not correct. Emend the figure indication.

Figure 2 D, E: you need indicate calibration bar, not magnification.

Figure 5A: the details are not clearly visible. You should increase the contrast and indicate better the several profiles. Merge what?

Discussion

Pag. 8, line 203-220: The paragraph is a summary of your work. It can be used to improve the final paragraph of discussion. In the actual position is rather confounding.

 Materials and Methods

Pag. 10, line 316: “…group were daily…” what? Correct and rephrase the sentence.

Reviewer 4 Report

Comments and Suggestions for Authors

The paper is interesting and up-to-date. However, I find a problem which may shed doubts when interpreting the results. The authors used C57BL/6 mice (6 weeks old) in the observation of induced vitiligo and its treatment. The important aspect of this study is the hair cycle. The authors have not mentioned the way of monitoring the hair cycle. The used animal model is widely used in the study of hair cycling. The age (6 weeks) is in this model the time of entering another anagen and developing a new round of hair growth. In these animals melanogenesis is strongly coupled with anagen. Melanocytes are here developed only in hair follicles, moreover, melanocytes are here undergoing a new melanogenesis (melanocyte generation) from appropriate stem cells. This process undergoes then (after ca. 16 days) a massive apoptosis and melanoinvolution. The telogen phase, quite a long one, is related to a total lack of melanin in skin and in hair follicles (the pelage hair, their root in telogen is also devoid of melanin). Therefore - this is a strong factor affecting the described results and must not be omitted in the interpretation, contrarywise, especially considered.

Another fact is that melanocytes and in general hair follicles posess a neuronal origin, and thus undergo an evident influence by the neural activity. This also must be reflected in the interpretation. Finally, vitiligo and alopecia areata are interdependent phenomena (see e.g papers by prof. Ralf Paus or Angela Chrisstiano). The dependence of the vagus nerve action in these observation is a new phenomenon, but the described facts shoul;d be obligatorily considered.

The number, date, and the name of ethical commission (council) who assigned the permission (as the experiments in vivo are invasive), should be disclosed explicitly in the text.

Comments on the Quality of English Language

There are several errors in English. Please have a double-check. E.g. the saxon genetive is overused (line 21 "the study's conclusions").

Round 2

Reviewer 2 Report

Comments and Suggestions for Authors

The manuscript by Luo, S. et al. describing a possible treatment of vitiligo in a mouse model through vagus nerve stimulation addressed all my suggestions. Moreover, I found the rebuttal of the authors to my comments really detailed. However, there is still an issue with the images. I think it can be solved by enlarging some images and rearranging the figures (see minor issues).

The language is fine and I can´t find relevant issues.

There are various minor comments:

- Results, Figure 2. It´s true you improved the image resolution, but it is still difficult to see the details. Why don´t you just enlarge the images? You can change the position of the different sections in the image to make the figure much bigger. This way, you can make B, D and E go from side to side improving their detail. You can even separate Figure 2 into 2 different figures.

- Results, Figure 4. I have the same comment for this figure. You should make D and E bigger. The easiest way is enlarging figures D and E across all the wideness of the page.

- Results, Figure 5. The same for this image. You should enlarge Figure 5A from side to side, but I think you should also enlarge the cytometry images.

- Results, Figure 5. Again, you should explain the meaning of the arrows in 5A. At this magnification and resolution, I can see nothing in this image. Nothing.

- Discussion, page 8, lines 212-248. This is an extremely long paragraph. I think you can divide it in line 228, where the text is highlighted in yellow.

- Materials and methods, page 9, line 292. I previously asked the number of mice because you should include this information at the beginning of Materials and Methods. You can add a line saying: 12 mice were purchased … and farmed … All the mice were male?

- Materials and methods, page 10, lines 321-323. You employed a confusing way of describing a quite simple procedure. What about this? Mice were divided into 3 groups composed of 4 individuals: (i) untreated vitiligo, (ii)  taVNS treated vitiligo, (iii) control.

Reviewer 4 Report

Comments and Suggestions for Authors

The manuscript is now ready for publication.

Author Response

We thank the reviewer for the comments.